

# Effectiveness and adequacy of blinding in the moderation of pain outcomes: Systematic review and meta-analyses of dry needling trials

Felicity A. Braithwaite[1,2], Julie L. Walters[1], Lok Sze Katrina Li[1], G. Lorimer Moseley[1,2], Marie T. Williams[1,3] and Maureen P. McEvoy[1]

[1] School of Health Sciences, University of South Australia, Adelaide, Australia
[2] Body in Mind research group, University of South Australia, Adelaide, Australia
[3] Alliance for Research in Exercise, Nutrition and Activity (ARENA), University of South Australia, Adelaide, Australia

Corresponding author
Felicity A. Braithwaite,
felicity.braithwaite@gmail.com

## ABSTRACT

**Background**. Blinding is critical to clinical trials because it allows for separation of specific intervention effects from bias, by equalising all factors between groups except for the proposed mechanism of action. Absent or inadequate blinding in clinical trials has consistently been shown in large meta-analyses to result in overestimation of intervention effects. Blinding in dry needling trials, particularly blinding of participants and therapists, is a practical challenge; therefore, specific effects of dry needling have yet to be determined. Despite this, dry needling is widely used by health practitioners internationally for the treatment of pain. This review presents the first empirical account of the influence of blinding on intervention effect estimates in dry needling trials. The aim of this systematic review was to determine whether participant beliefs about group allocation relative to actual allocation (blinding effectiveness), and/or adequacy of blinding procedures, moderated pain outcomes in dry needling trials.

**Methods**. Twelve databases (MEDLINE, EMBASE, AMED, Scopus, CINAHL, PEDro, The Cochrane Library, Trove, ProQuest, trial registries) were searched from inception to February 2016. Trials that compared active dry needling with a sham that simulated dry needling were included. Two independent reviewers performed screening, data extraction, and critical appraisal. Available blinding effectiveness data were converted to a blinding index, a quantitative measurement of blinding, and meta-regression was used to investigate the influence of the blinding index on pain. Adequacy of blinding procedures was based on critical appraisal, and subgroup meta-analyses were used to investigate the influence of blinding adequacy on pain. Meta-analytical techniques used inverse-variance random-effects models.

**Results**. The search identified 4,894 individual publications with 24 eligible for inclusion in the quantitative syntheses. In 19 trials risk of methodological bias was high or unclear. Five trials were adequately blinded, and blinding was assessed and sufficiently reported to compute the blinding index in 10 trials. There was no evidence of a moderating effect of blinding index on pain. For short-term and long-term pain assessments pooled effects for inadequately blinded trials were statistically significant in favour of active dry needling, whereas there was no evidence of a difference between active and sham groups for adequately blinded trials.

![PeerJ logo]

**Discussion**. The small number and size of included trials meant there was insufficient evidence to conclusively determine if a moderating effect of blinding effectiveness or adequacy existed. However, with the caveats of small sample size, generally unclear risk of bias, statistical heterogeneity, potential publication bias, and the limitations of subgroup analyses, the available evidence suggests that inadequate blinding procedures could lead to exaggerated intervention effects in dry needling trials.

## BACKGROUND

Blinding is widely considered critical to the internal validity of clinical trials because it allows separation of specific intervention effects from effects due to bias. This separation is possible because blinding equalises all factors between groups except for the proposed mechanism of action of the intervention under investigation (*Hróbjartsson et al., 2014*).

Blinding adequacy relates to procedures in the design of a trial to blind relevant parties (i.e., trial staff, therapists, recipients, outcome assessors, data analysts). In the absence of adequate procedures, the inclination of these parties to favour a particular result can lead to distorted findings that most commonly manifest as exaggerated intervention effects (*Hróbjartsson et al., 2014*; *Hróbjartsson et al., 2012*; *Hróbjartsson et al., 2013*; *Jüni, Altman & Egger, 2001*; *Moher et al., 2010*; *Nüesch et al., 2009*; *Savović et al., 2012*; *Schulz et al., 1995*; *Wood et al., 2008*). For example, a review of trials involving head-to-head comparisons of blinded versus non-blinded participants demonstrated pronounced bias in non-blinded groups for complementary/alterative interventions ($N = 12$ trials, 11 of which were acupuncture trials) (*Hróbjartsson et al., 2014*). Self-reported outcomes such as pain, which is often used to evaluate physical interventions, are particularly susceptible to the effects of inadequate blinding procedures (*Hróbjartsson et al., 2014*; *Hróbjartsson et al., 2013*; *Moher et al., 2010*; *Savović et al., 2012*; *Wood et al., 2008*). The complex nature of physical interventions means that blinding of relevant parties, particularly participants and therapists, is often extremely difficult (*Boutron et al., 2004*). As a result, blinding procedures for these types of interventions have been generally inadequate or omitted completely (*Armijo-Olivo et al., 2017*; *Boutron et al., 2007*; *Boutron et al., 2004*; *Machado et al., 2008*; *Moseley et al., 2011*).

Inclusion of adequate blinding procedures is recognised as crucial to robust trial design (*Moher et al., 2010*), but evaluation of the actual effectiveness of blinding procedures and its influence on clinical trial outcomes has been poorly addressed (*Bang et al., 2010*; *Fergusson et al., 2004*; *Hróbjartsson et al., 2007*). Needling therapies [acupuncture and dry needling (DN)] provide a unique intervention on which to focus evaluation of blinding effectiveness, because unlike trials of many other physical interventions (*Armijo-Olivo et al., 2017*; *Hróbjartsson et al., 2007*; *Machado et al., 2008*; *Villamar et al., 2013*), blinding assessments are becoming common practice in needling therapy trials (*Moroz et al., 2013*).
In addition, needling therapies are growing globally in popularity to manage pain (*Cagnie et al., 2013*; *Carlesso et al., 2014*; *Dommerholt, 2011*; *Legge, 2014*) and encompass a range of factors known to predict large non-specific responses; needling therapies are ritualistic, invasive, involve a medical device, are highly credible to patients, and are often held in high regard by the person delivering them (*Benedetti, 2013*; *Finniss et al., 2010*; *Kaptchuk, 2002*; *Kaptchuk et al., 2008*; *Kaptchuk & Miller, 2015*; *Kaptchuk et al., 2006*). Exaggeration of intervention effects in acupuncture is associated with expectation of intervention outcomes (*Colagiuri & Smith, 2011*; *Linde et al., 2007*), and in acupuncture trials, beliefs about group allocation have been shown to bear a stronger relationship to pain than actual allocation (*Bausell et al., 2005*; *Vase et al., 2013*; *White et al., 2012*). These findings suggest that failed blinding could be a significant confounder of trial outcomes, and confirms that well-blinded trials will be required to determine the mechanisms of needling therapies. However, a recent systematic review of acupuncture and dry needling trials ($N = 54$ trials) reported that only 61% of trials *might* have had effective participant blinding based on empirical data (i.e., where participant beliefs about the intervention to which they were allocated were approximately balanced between active and sham groups) (*Moroz et al., 2013*). Ineffective participant blinding, coupled with potentially inadequate or omitted blinding procedures for other relevant parties (particularly therapists), calls into question any specific intervention effect of needling therapy reported to date.

This systematic review presents the first empirical account of the influence of blinding on intervention effect estimates in dry needling trials. Dry needling differs from acupuncture because while acupuncture needles are used, they are inserted into clinically identified locations in muscles (such as tender areas, palpable nodules or bands) rather than the largely pre-determined insertion sites based on traditional Chinese medicine used in acupuncture. As such, dry needling aims at local effects whereas acupuncture aims at systemic effects. The aim of this review was to determine the influence of blinding effectiveness and blinding adequacy on pain in sham-controlled dry needling trials. Blinding effectiveness was determined by participant beliefs about group allocation relative to actual allocation, and blinding adequacy was determined by critical appraisal. This review posed two questions: (1) 'Does blinding effectiveness moderate intervention effect on pain?' and (2) 'Does blinding adequacy moderate intervention effect on pain?'

## METHODS

The methods complied with the Preferred Reporting Items for Systematic Reviews and Meta-Analyses (PRISMA) checklist (*Moher et al., 2009*). The protocol was prospectively registered with the International Prospective Register of Systematic Reviews (PROSPERO) (registration number: 42016029340; URL: http://www.crd.york.ac.uk/prospero/display_record.asp?ID=CRD42016029340).

### Disclosure of deviations from prospectively registered protocol

Following original registration on March 2, 2016, two changes were made to the protocol of this review: (1) the data extraction template was pilot tested using an iterative process rather than a sample of 10 included trials, and percentage agreement was used to determine

**Table 1  MEDLINE search strategy.**

| Search terms | | Limits applied |
|---|---|---|
| 1. | Needl*.tw | Humans only |
| 2. | *Acupuncture therapy/ | Keyword searches limited to title/abstract/keyword fields |
| 3. | Acupuncture.tw | |
| 4. | Intramuscular stimulation.tw | |
| 5. | Sham.tw | |
| 6. | *Placebo effect/ | |
| 7. | *Placebos/ | |
| 8. | Placebo$1.tw | |
| 9. | #1 OR #2 OR #3 OR #4 | |
| 10. | #5 OR #6 OR #7 OR #8 | |
| 11. | #9 AND #10 | |

agreement, rather than an intra-class correlation coefficient (ICC); and (2) the time-point that was used to investigate the influence of blinding effectiveness on pain outcomes was the time-point at which blinding was assessed (instead of the pre-defined time-points of immediate, short-term, and long-term pain assessments), because the authors agreed that this time-point would most accurately reflect intervention beliefs (i.e., blinding effectiveness) as beliefs can change over time (*Bang et al., 2010*; *Rees et al., 2005*). The second change was updated in PROSPERO prior to data analyses (revision posted on February 5, 2017). This review presents only review questions 1 and 2 of the protocol; review questions 3 and 4 will be reported elsewhere.

## Information sources and search strategy

One reviewer (FAB) executed the search strategy. Databases (MEDLINE, EMBASE, AMED, Scopus, CINAHL, PEDro, The Cochrane Library) were searched from inception to February 2016. The general search terms were (needl* OR acupuncture OR intramuscular stimulation) AND (sham OR placebo*), and Medical Subject Headings (MeSH) were used where possible. The full electronic search strategy for MEDLINE is presented in Table 1. Searches were modified to suit the functionality of each database. Thesis databases (Trove, ProQuest) and clinical trial registries (Australian New Zealand Clinical Trials Registry (ANZCTR), Clinicaltrials.gov, World Health Organization International Clinical Trials Registry Platform (WHO ICTRP)) were crosschecked with database searches to identify further potential trials. The reference lists of systematic reviews identified by the search were examined to locate additional or unpublished trials. There were no limits on year, language, or publication status.

## Eligibility and study selection

Trials were eligible for inclusion in this review if they (1) were prospective experimental designs (e.g., randomised, non/quasi-randomised trials, pre-post, n-of-1) of any duration, which included a 'real' dry needling intervention (referred to as 'active' dry needling in this review) and a placebo/sham dry needling intervention; (2) included human adults ($\geq$18 years of age) who were asymptomatic or with symptomatic health conditions; (3) involved

a recognised dry needling approach with needle insertion sites based on anatomical or clinical rationales; (4) assessed and reported an outcome for pain [visual analogue scale (VAS) or numeric rating scale (NRS)]. Trials were also eligible for inclusion if they reported blinding assessment data, without reporting on pain, but the results from these trials are not presented in this review. Trials were ineligible for inclusion if the needling therapy involved pre-designated needle insertion sites (e.g., traditional acupuncture points) or involved injection of a substance (wet needling).

Records identified from the search strategy were exported to Endnote, duplicates were removed, and the remaining records were imported into the online screening tool 'Covidence systematic review software' (*Anonymous, 2018*). Titles and abstracts were screened against the eligibility criteria by three independent reviewers in teams of two (FAB and MPM or JLW), and trials potentially meeting the criteria were progressed to full text review. The same three reviewers independently screened the full-text articles in teams of two. Discrepancies were resolved through discussion, with an independent third reviewer (MPM, JLW, or LSKL) consulted where necessary. Where full-text was unavailable, authors were contacted to clarify eligibility and/or to provide full-texts. Non-English publications were translated using Google Translate; the extracted data were then checked with fluent speakers of each language.

## Data extraction and Risk of Bias (RoB) assessment

A prospectively designed data extraction template was developed based on the Standards for Reporting Interventions in Controlled Trials of Acupuncture (STRICTA) (*MacPherson et al., 2010*) and the Cochrane Handbook 'Checklist of items to consider in data collection or data extraction' (*Higgins & Green, 2011*). The domains of data extraction were: source details, trial demographics, trial design, participant details, therapist details, intervention details, outcomes (pain and blinding assessment), blinding strategies, sample size and dropouts, results (pain and blinding assessment), and key conclusions of the authors.

The provisional data extraction template was pilot tested for inter-rater agreement by two reviewers (FAB and JLW) using an iterative process (two randomly selected included trials in each iteration). Once the pre-specified level of inter-rater agreement was established ($\geq$75% agreement of items within an individual trial), two independent reviewers performed the remaining data extraction (FAB and LSKL, JLW, or MPM), with a third reviewer consulted to resolve disagreements as required.

Only data from the first phase of crossover trials were extracted due to the risk of carry-over intervention effects. Where necessary (i.e., where no text or table data were provided), graphical data were extracted using a ruler; if there were differences in these values between the two extracting reviewers, the average value was calculated. Pain intensity data were converted to a 100-point continuous scale where required (e.g., if collected using a 10 cm VAS or an NRS).

Risk of Bias (RoB) of individual trials was assessed using the Cochrane RoB assessment tool for randomised trials (because all included studies were randomised trials) (*Higgins et al., 2011*). Three key domains (allocation concealment, performance bias, detection bias) were determined *a priori* based on relevance to the review questions. The key domains

were informed by empirical evidence for the likelihood and magnitude of these biases influencing trial outcomes (*Higgins et al., 2011*; *Hróbjartsson et al., 2014*; *Hróbjartsson et al., 2013*; *Savović et al., 2012*; *Wood et al., 2008*). The overall RoB for individual trials was determined using the three key domains (low = low RoB for all key domains, unclear = low or unclear RoB for all key domains, high = high RoB for one or more key domains) (*Higgins et al., 2011*). Two independent reviewers appraised each trial (FAB and MPM, JLW, or LSKL), with a third reviewer consulted to resolve disagreements as required.

Publication bias for each meta-analysis was assessed by visual inspection of asymmetry of funnel plots, which were contour-enhanced to allow consideration of the potential influence of the statistical significance of trial outcomes on publication bias (*Peters et al., 2008*). A statistical test for asymmetry was also computed for funnel plots containing ≥10 trials using the method specified in *Egger et al. (1997)* at a significance level of $p < 0.10$ (*Higgins & Green, 2011*; *Sterne et al., 2011*).

## Data syntheses

For both review questions, meta-analyses used generic inverse variance and random-effects models. Restricted Maximum Likelihood (REML) was used to estimate between-trial variance. Stata statistical software (version 15.1) (*StataCorp, 2017*) was used to compute inferential statistics and create plots. The $x^2$ test and $I^2$ statistic were used to assess statistical heterogeneity; $p < 0.10$ was interpreted as statistically significant heterogeneity and $I^2 > 50\%$ was interpreted as substantial heterogeneity (*Higgins & Green, 2011*). Intervention effects were interpreted as statistically significant when $p < 0.05$, and between-group effect sizes [Standardised Mean Difference (SMD)] were considered large if $>0.80$, moderate if between 0.20 and 0.80, and small if $<0.20$ as defined by *Cohen (1988)*.

A blinding index (BI) (*Bang, Ni & Davis, 2004*) was used to quantify the effectiveness of blinding (participant belief about group allocation relative to actual group allocation), where blinding assessments were sufficiently reported. The BI estimates the degree of unblinding (i.e., correct identification of group allocation) beyond random chance (*Bang, Ni & Davis, 2004*). To assist with interpretation of blinding effectiveness, groups within included trials were classified based on the BI cut-offs proposed by *Moroz et al. (2013)* (Table 2). Trials were then categorised based on paired classifications for the active and sham groups, termed a 'blinding scenario' (e.g., 'Correct/Incorrect', which means that the active group was classified as 'Correct' and the sham group was classified as 'Incorrect') (*Bang et al., 2010*). Using this classification method, a total of nine blinding scenarios were possible (*Bang et al., 2010*). The 'R' software package (version 3.4.3) (*R Core Team, 2017*) was used to compute BIs and their 95% Confidence Intervals (CIs).

### Review question 1: Does blinding effectiveness moderate intervention effect on pain?

It was hypothesised that if the proportion of participants who believed they had the active or sham intervention differed between active and sham groups (i.e., unbalanced intervention beliefs), this would have a moderating effect on between-group pain outcomes (i.e., increase or decrease between-group differences). To interrogate the hypothesis, a summary value for blinding effectiveness for each trial was calculated by adding the BI scores from each

**Table 2** Interpretation of the Blinding Index (BI) and classifications.

| BI | Interpretation | BI cut-offs[a] | Classification |
|---|---|---|---|
| −1.00 | All participants mistakenly guess the alternative intervention (incorrect guessing) | $BI \leq -0.20$ | Incorrect |
| 0.00 | Random guessing (ideal blinding) | $-0.20 < BI < 0.20$ | Random |
| +1.00 | All participants correctly guess their allocation (correct guessing) | $BI \geq 0.20$ | Correct |

Notes.
[a]Cut-off scores were developed by consensus of authors of *Moroz et al. (2013)* and should not be interpreted as definitive classifications of blinding effectiveness.
BI, Blinding Index.

group (i.e., BI active group + BI sham group) (adapted from *Freed et al. (2014)*), and a meta-regression of the influence of the summary BI (blinding effectiveness) on between-group effect size (pain) was computed for the time-point closest to which blinding was assessed (as this is likely to most accurately reflect intervention beliefs at that moment).

### Review question 2: Does blinding adequacy moderate intervention effect on pain?

The Cochrane RoB tool (*Higgins et al., 2011*) was also used to assess blinding adequacy of trials. Adequacy was based on the four RoB domains that relate to blinding (allocation concealment, participant blinding, therapist blinding, and outcome assessor blinding) (*Higgins et al., 2011*) (adapted from *Feys et al. (2014)*). Trials were conservatively categorised as either '*adequately blinded*' or '*inadequately blinded*' based on the following rules:

- Adequately blinded: low RoB across all four domains, or low RoB in the three domains excluding 'therapist blinding' if no trials attempted therapist blinding.
- Inadequately blinded: high or unclear RoB in at least one domain.

Meta-analyses were used to assess differences in between-group effect sizes based on adequacy of blinding. It was hypothesised that inadequate blinding would favour active dry needling. Separate meta-analyses were completed for three time-periods: immediately after the first/only intervention (<24 h); short-term (24 h to one month from baseline, using closest assessment to one week); long-term (one to six months from baseline, using closest assessment to three months).

## RESULTS

### Outcome of search strategy

The outcome of the search strategy is presented in Fig. 1. The search identified 11835 records. Four additional publications were identified by searching personal records (*Itoh, Katsumi & Kitakoji, 2004*) and through hand searching reference lists of 199 systematic reviews (*Itoh & Katsumi, 2005*; *Itoh et al., 2006b*; *Katsumi et al., 2004*). Following removal of duplicates, 4894 potentially relevant publications were screened. Title and abstract screening excluded 4280 publications. Of the remaining 614 publications, 588 were excluded following full-text review, leaving 26 publications (Fig. 1). The exclusion of two
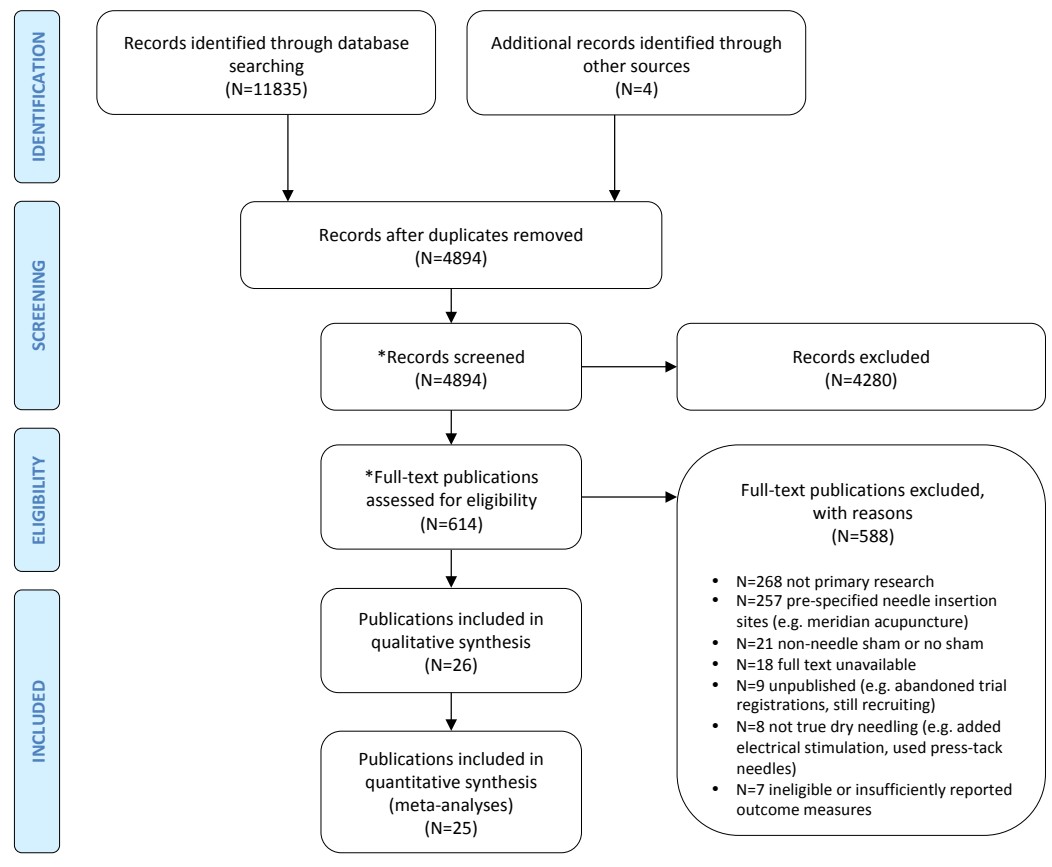

**Figure 1** **Preferred Reporting Items for Systematic Reviews and Meta-Analyses (PRISMA) flow diagram (http://www.prisma-statement.org).** *Processes performed by two independent reviewers.

research questions from the current review resulted in the exclusion of three trials (within one publication) from this review because they did not report a pain outcome (*Braithwaite, 2014*) (this publication is included in Fig. 1 because the two omitted review questions that did include results from this publication are reported elsewhere). The 25 relevant publications included one trial that presented results over two publications (*Tough et al., 2010*; *Tough et al., 2009*), and two single publications with two eligible sham groups (*Itoh & Katsumi, 2005*; *Itoh et al., 2007*); therefore, 25 publications (with 26 group comparisons from 24 trials) are presented in the current review. Of these 25 publications, 24 publications (with 25 group comparisons from 23 trials) provided sufficient data for inclusion in the current meta-analyses. For the meta-analyses, in the two trials with two eligible sham groups (*Itoh & Katsumi, 2005*; *Itoh et al., 2007*) the active group data were used twice.

Five non-English publications were included in the current three Japanese publications (*Itoh & Katsumi, 2005*; *Itoh et al., 2006b*; *Katsumi et al., 2004*) and two Spanish publications (*Espejo Antúnez et al., 2014*; *García-Gallego et al., 2011*).

Of 22 authors who were contacted to clarify eligibility and/or to provide full-texts, 12 replied confirming ineligibility and 10 did not reply. For eight further records, author

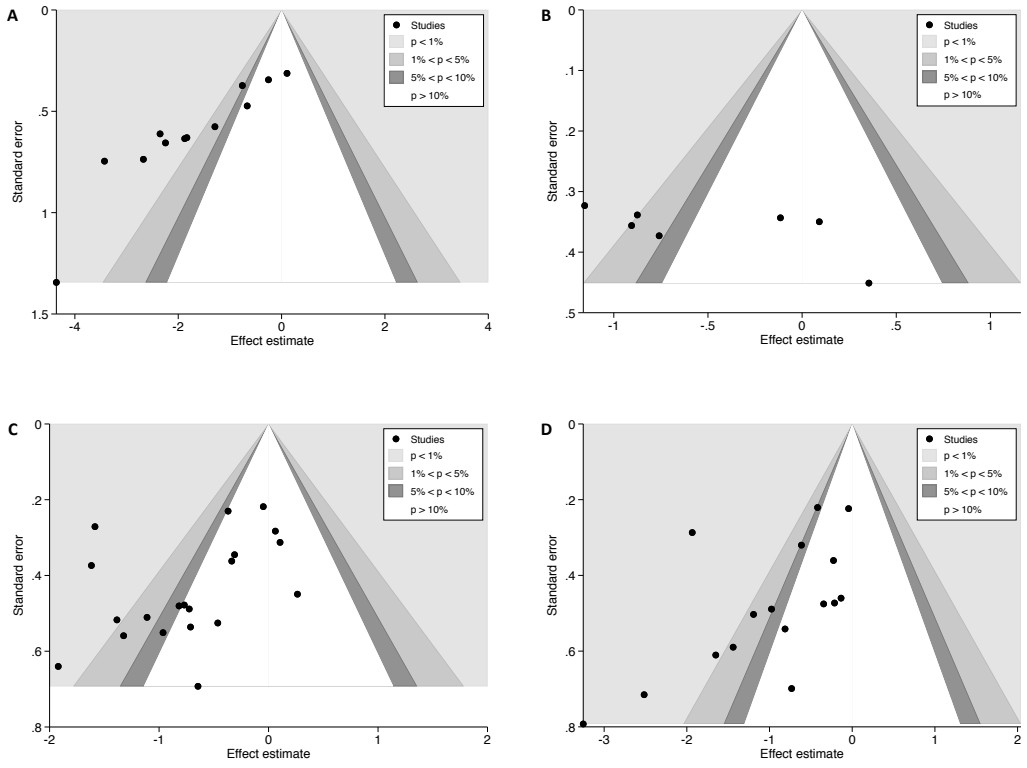

**Figure 2  Contour-enhanced funnel plots for pain outcomes in dry needling trials.** (A) Funnel plot for Review question 1 (blinding effectiveness): time-point closest to when blinding was assessed. (B, C, D) Funnel plots for Review question 2 (blinding adequacy) ((B) Pain assessments immediately after first/only intervention; (C) short-term pain assessments; (D) long-term pain assessments).

contact details could not be found. Nine authors of included trials were contacted to clarify trial details or request data; one author replied stating they no longer had access to the data, and the remaining eight authors did not reply.

## Risk of Bias (RoB) assessment

A summary of results for the RoB assessment is presented in Table 3. Overall RoB was high in one trial, unclear in 18 trials, and low in five trials (Table 3). The areas with least RoB were participant blinding and reporting bias (low RoB in all included trials). The areas with greatest potential for bias were blinding of therapists and research personnel (high or unclear RoB in all included trials), allocation concealment, and attrition bias (Table 3).

### *Assessment of publication bias*

Visual inspection of asymmetry of contour-enhanced funnel plots suggested that publication bias may be present (Fig. 2) (*Peters et al., 2008*). A statistical test for asymmetry was computed for funnel plots containing ≥10 group comparisons (Figs. 2A, 2C and 2D) and a statistically significant result was found for all three plots ($p < 0.001$, $p = 0.083$, and $p = 0.061$, respectively), which further supports the presence of publication bias (*Egger et al., 1997*).

**Table 3  Risk of bias assessment (*N* = 24 trials) (Cochrane Risk of Bias tool for randomised trials (*Higgins et al. 2011*)).**

| Author & year | Random allocation | [ab]Allocation concealed | Performance bias | | | | [ab]Detection bias (OAB) | Attrition bias | Reporting bias | OVERALL | Adequately blinded? |
| --- | --- | --- | --- | --- | --- | --- | --- | --- | --- | --- | --- |
| | | | [b]PB | RPB | [b]TB | [a]Overall | | | | | |
| *Cotchett, Munteanu & Landorf (2014)* | ✓ | ✓ | ✓ | ? | × | ✓ | ✓ | ✓ | ✓ | ✓ | Adequate |
| *Dıraçoğlu et al. (2012)* | ✓ | ? | ✓ | ? | × | ✓ | ✓ | × | ✓ | ? | Inadequate |
| *Espejo Antúnez et al. (2014)* | ✓ | ? | ✓ | ? | × | ✓ | ✓ | ✓ | ✓ | ? | Inadequate |
| *García-Gallego et al. (2011)* | ✓ | ? | ✓ | ? | × | ✓ | ✓ | ✓ | ✓ | ? | Inadequate |
| *Huguenin et al. (2005)* | ✓ | ✓ | ✓ | ? | × | ✓ | ✓ | × | ✓ | ✓ | Adequate |
| *Inoue et al. (2006)* | ✓ | ✓ | ✓ | ? | × | ✓ | ✓ | ✓ | ✓ | ✓ | Adequate |
| *Itoh, Katsumi & Kitakoji (2004)* | ✓ | ? | ✓ | ? | × | ✓ | ✓ | × | ✓ | ? | Inadequate |
| *Itoh & Katsumi (2005)* | ✓ | ✓ | ✓ | ? | × | ✓ | ✓ | × | ✓ | ✓ | Adequate |
| *Itoh et al. (2006a)* | ✓ | ? | ✓ | ? | × | ✓ | ✓ | × | ✓ | ? | Inadequate |
| *Itoh et al. (2006b)* | ? | ? | ✓ | ? | × | ✓ | ? | × | ✓ | ? | Inadequate |
| *Itoh et al. (2007)* | ✓ | ? | ✓ | ? | × | ✓ | ✓ | × | ✓ | ? | Inadequate |
| *Itoh et al. (2008)* | ✓ | ? | ✓ | ? | × | ✓ | ✓ | × | ✓ | ? | Inadequate |
| *Itoh et al. (2012)* | ✓ | ? | ✓ | ? | × | ✓ | ✓ | × | ✓ | ? | Inadequate |
| *Itoh et al. (2014)* | ✓ | ? | ✓ | ? | × | ✓ | ✓ | × | ✓ | ? | Inadequate |
| *Katsumi et al. (2004)* | ✓ | ? | ✓ | ? | × | ✓ | ? | ✓ | ✓ | ? | Inadequate |
| *Mayoral et al. (2013)* | ✓ | ? | ✓ | ? | × | ✓ | ✓ | × | ✓ | ? | Inadequate |
| *McMillan, Nolan & Kelly (1997)* | ? | ? | ✓ | ? | × | ✓ | ✓ | ? | ✓ | ? | Inadequate |
| *Myburgh et al. (2012)* | ✓ | ? | ✓ | ? | × | ✓ | ? | × | ✓ | ? | Inadequate |
| *Nabeta & Kawakita (2002)* | ✓ | ? | ✓ | ? | × | ✓ | ? | ✓ | ✓ | ? | Inadequate |
| *Pecos-Martín et al. (2015)* | ✓ | ? | ✓ | ? | × | ✓ | ✓ | ✓ | ✓ | ? | Inadequate |
| *Sterling et al. (2015)* | ✓ | ✓ | ✓ | ? | × | ✓ | ✓ | ✓ | ✓ | ✓ | Adequate |
| *Tekin et al. (2013)* | ✓ | ? | ✓ | ? | × | ✓ | ✓ | × | ✓ | ? | Inadequate |
| *Tough et al. (2009)/ Tough et al. (2010)* | ✓ | ? | ✓ | × | × | × | ? | ✓ | ✓ | × | Inadequate |
| *Tsai et al. (2010)* | ✓ | ? | ✓ | ? | × | ✓ | ✓ | ✓ | ✓ | ? | Inadequate |

**Notes.**

[a]Key domains (used to determine overall Risk of Bias for individual trials).

[b]Domains used to determine blinding adequacy.

PB, Participant Blinding; RPB, Research Personnel Blinding; TB, Therapist Blinding; OAB, Outcome Assessor Blinding; ✓, Low RoB; ?, Unclear RoB; ×, High RoB.

## Description of included trials

Table 4 presents a summary of trial characteristics and results of the 24 trials involving 26 group comparisons. Non-penetrating (NP) shams were used in 16 group comparisons, penetrating (P) shams were used in nine group comparisons, and one group comparison used anaesthesia (general or spinal) to blind participants (*Mayoral et al., 2013*) (Table 4). The 16 NP devices were guide-tubes alone ($N = 3$ group comparisons), custom-made blunted/retracting needles ($N = 12$ group comparisons), and one commercial device (the Park sham *Park et al., 1999*). Of the nine group comparisons that used penetrating shams, six inserted needles subcutaneously only (i.e., superficial dry needling above trigger points (TrP SDN) or away from trigger points (Non-TrP SDN)), and three inserted needles into muscle but away from trigger points (Non-TrP DN) (Table 4).

Fourteen trials (16 group comparisons) assessed blinding effectiveness ($N = 13$ trials) or intervention credibility ($N = 1$ trial). To assess blinding effectiveness, participants were asked whether they thought a needle had been inserted ($N = 9$ trials), if they felt a 'needling sensation' ($N = 1$ trial), or which group they thought they were in ($N = 1$ trial). The remaining two trials did not report how blinding was assessed (*Huguenin et al., 2005*; *Sterling et al., 2015*). To assess intervention credibility, participants completed the Credibility/Expectancy Questionnaire (CEQ) (*Devilly & Borkovec, 2000*) ($N = 1$ trial). Of the 13 trials that assessed blinding effectiveness, 10 trials (12 group comparisons) presented blinding data in a way that the BI could be calculated for active and sham groups (Table 4). To evaluate blinding effectiveness, nine trials used inferential statistics to determine if there was a difference in proportions of guesses, and four trials described, but did not statistically analyse, the blinding data (Table 4). For pain outcome assessments, 22 trials used a VAS ($N = 11$ trials used a 100 mm scale and $N = 11$ trials used a 10 cm scale) and two trials used an NRS (NRS 0-10).

## Data syntheses

### Review question 1: Does blinding effectiveness moderate intervention effect on pain?

Figure 3 presents a bubble plot (meta-regression) of the influence of the summary BI on effect size (pain). There was no evidence of a moderating effect of the summary BI on effect size (meta-regression coefficient $-1.87$ (95% CI [$-5.63$–$1.88$]); $p = 0.292$; $N = 12$ group comparisons; $n = 248$) (Fig. 3). There was evidence of statistically significant and substantial statistical heterogeneity ($I^2 = 79.0\%$; $p < 0.001$) (*Higgins & Green, 2011*).

### Review question 2: Does blinding adequacy moderate intervention effect on pain?

Five of the 24 trials were adequately blinded (Table 3). All trials demonstrated adequate participant blinding and no trials attempted to blind therapists, so by default blinding adequacy was determined based on the remaining two domains (allocation concealment and blinding of outcome assessors).

Braithwaite et al. (2018), *PeerJ*, DOI 10.7717/peerj.5318

**Table 4  Characteristics and results of included group comparisons ($N = 26$ group comparisons).**

| Author & year | n | Dropouts[a] [reasons] | Type of sham | Blinding index (95% CI) Blinding scenario (AG/SG) | Reported blinding results | Reported blinding conclusion | Between-group SMD (pain) and reported p-values [−ve values in favour of AG] |
|---|---|---|---|---|---|---|---|
| *Cotchett, Munteanu & Landorf (2014)* | 84 | 5<br>AG: 3 [1 missed Ax; 2 ceased Ix]<br>SG: 2 [1 missed Ax; 1 ceased Ix] | NP: Custom (blunt needle) | Insufficient data | NSD between groups (CEQ) ($p > 0.05$ for all questions) | Success | ST: $-0.05$ ($p = 0.026$)<br>LT: $-0.42$ ($p = 0.007$) |
| *Huguenin et al. (2005)* | 52 | 7 [difficulty attending] | NP: Custom (blunt needle) | Insufficient data | AG only:<br>• Immed: Correct ($p = 0.001$)<br>• ST: NSD between correct and incorrect guesses ($p = 0.062$) | Success | Immed/B: [d] (NSD)<br>ST: [d] (NSD) |
| *Inoue et al. (2006)* | 31 | 0 | NP: Custom (guide tube only) | AG: 0.20 ($-0.30$–0.70)<br>SG: 0.25 ($-0.22$–0.72)<br>Correct/Correct | NSD between groups (p NR)<br>AG: 9/15 correct<br>SG: 10/16 correct | Success | Immed/B: 0.76 ($p = 0.020$) |
| [b]*Itoh & Katsumi (2005)* (NP) | 19 | 3<br>AG: 1<br>SG: 2<br>[All groups: 5 DNR; 2 AE] | NP: Custom (guide tube only) | [c]AG: 0.60 (0.10–1.10)<br>[c]SG: $-0.11$ ($-0.76$–0.54)<br>Correct/Random | NSD between groups ($p = 0.64$)<br>AG: 8/10 correct<br>SG: 4/9 correct | Success | ST: $-0.82$ ($p < 0.05$)<br>B: $-2.35$ ($p < 0.01$)<br>LT: $-0.98$ (NSD) |
| [b]*Itoh & Katsumi (2005)* (P) | 19 | 3<br>AG: 1<br>SG: 2<br>[All groups: 5 DNR; 2 AE] | P: TrP SDN | [c]AG: 0.60 (0.10–1.10)<br>[c]SG: $-0.20$ ($-0.81$–0.41)<br>Correct/Incorrect | NSD between groups ($p = 0.64$)<br>AG: 8/10 correct<br>SG: 4/10 correct | Success | ST: $-0.77$ (NSD)<br>B: $-0.67$ ($p < 0.05$)<br>LT: $-0.13$ (NSD) |
| *Sterling et al. (2015)* | 80 | 7<br>AG: 3 [LTFU]<br>SG: 4 [LTFU] | NP: Commercial (Park sham) | Insufficient data | Descriptive only<br>SG: 1/36 correct<br>[All remaining participants believed AG or DK] | Success | LT: $-0.04$ (NSD)<br>B: $-0.09$ (NSD) |
| *Dıraçoğlu et al. (2012)* | 50 | 2<br>AG: 1 [difficulty attending]<br>SG: 1 [DNR] | P: Non-TrP SDN | Did not assess blinding | – | – | ST: 0.06 ($p = 0.478$) |
| *Espejo Antúnez et al. (2014)* | 45 | 0 | NP: Custom (retracting needle) | Did not assess blinding | – | – | Immed: $-1.15$ ($p < 0.01$) |

Braithwaite et al. (2018), *PeerJ*, DOI 10.7717/peerj.5318

| Author & year | n | Dropouts[a] [reasons] | Type of sham | Blinding index (95% CI) Blinding scenario (AG/SG) | Reported blinding results | Reported blinding conclusion | Between-group SMD (pain) and reported p-values [−ve values in favour of AG] |
|---|---|---|---|---|---|---|---|
| *García-Gallego et al. (2011)* | 33 | 0 | P: Non-TrP DN | Did not assess blinding | – | – | Immed: 0.09 (NSD) ST: 0.17 (NSD) |
| *Itoh, Katsumi & Kitakoji (2004)* | 18 | 4 AG: 1 [AE] SG: 3 [DNR] | P: TrP SDN | Did not assess blinding | – | – | ST: −0.72 (NSD) LT: −0.21 (NSD) |
| *Itoh et al. (2006a)* | 19 | 7 AG: 3 [2 DNR; 1 AE] SG: 4 [DNR] | NP: Custom (blunt needle) | AG: 0.50 (0.00–1.00) SG: −0.11 (−0.68–0.46) Correct/Random | NSD between groups ($p = 0.38$) AG: 7/10 correct; 1/10 DK SG: 3/9 correct; 2/9 DK | Success | ST: −1.38 (NSD) B: −3.43 ($p < 0.001$) LT: −1.19 (NSD) |
| *Itoh et al. (2006b)* | 18 | 5 AG: 2 SG: 3 [All groups: 4 DNR; 2 drugs] | NP: Custom (blunt needle) | AG: NR SG: −0.56 (−1.00−−0.11) NR/Incorrect | Descriptive only (SG only) SG: 1/9 correct; 2/9 DK | NR | ST/B: −1.11 ($p < 0.05$) LT: −0.35 (NSD) |
| [b] *Itoh et al. (2007)* (NP) | 15 | 5 AG: 2 [1 DNR; 1 AE] SG: 3 [2 DNR; 1 AE) | NP: Custom (blunt needle) | AG: 0.38 (−0.11–0.86) SG: −0.29 (−0.94–0.37) Correct/Incorrect | NSD between groups ($p = 0.89$) AG: 4/8 correct; 3/8 DK SG: 2/7 correct; 1/7 DK | Success | ST: −0.71 (NSD) B: −1.87 (NSD) LT: −2.52 (NSD) |
| [b] *Itoh et al. (2007)* (P) | 16 | 4 AG: 2 [1 DNR; 1 AE] SG: 2 [1 DNR; 1 AE] | P: Non-TrP DN | AG: 0.38 (−0.11–0.86) SG: −0.38 (−0.86–0.11) Correct/Incorrect | NSD between groups ($p = 0.89$) AG: 4/8 correct; 3/8 DK SG: 1/8 correct; 3/8 DK | Success | ST: −1.32 (NSD) B: −2.25 (NSD) LT: −3.25 (NSD) |
| *Itoh et al. (2008)* | 15 | 5 AG: 2 [1 DNR; 1 AE] SG: 3 [DNR] | NP: Custom (blunt needle) | AG: 0.75 (0.29–1.21) SG: −0.43 (−1.10–0.24) Correct/Incorrect | NSD between groups ($p = 0.74$) AG: 7/8 correct SG: 2/7 correct | Success | ST: −1.95 B: −2.67 LT: −0.81 (AUC $p = 0.025$) |
| *Itoh et al. (2012)* | 15 | 1 AG: 1 [AE] | NP: Custom (blunt needle) | AG: 1.00 (1.00–1.00) SG: −1.00 (−1.00−−1.00) Correct/Incorrect | Descriptive only [All participants believed they were in AG] | Success | ST: −0.46 B: −1.83 LT: −1.65 (AUC $p = 0.003$) |
| *Itoh et al. (2014)* | 15 | 1 SG: 1 [DNR] | NP: Custom (blunt needle) | AG: 0.56 (0.01–1.10) SG: −0.50 (−1.10–0.10) Correct/Incorrect | NSD between groups ($p = 0.89$) AG: 7/9 correct SG: 2/8 correct | Success | ST: −0.96 B: −1.29 LT: −1.44 (AUC $p = 0.024$) |
| *Katsumi et al. (2004)* | 9 | 0 | NP: Custom (guide tube only) | AG: 1.00 (1.00–1.00) SG: −0.60 (−1.30–0.10) Correct/Incorrect | Descriptive only AG: 4/4 correct SG: 1/5 correct | NR | ST: −0.64 (NR) B: −4.36 (NR) LT: −0.73 (NR) |

**Table 4** (*continued*)

| Author & year | n | Dropouts[a] [reasons] | Type of sham | Blinding index (95% CI) Blinding scenario (AG/SG) | Reported blinding results | Reported blinding conclusion | Between-group SMD (pain) and reported *p*-values [−ve values in favour of AG] |
|---|---|---|---|---|---|---|---|
| *Mayoral et al. (2013)* | 31 | 9 AG: 4 [LTFU] SG: 5 [LTFU] | No needle: GA/SA | Did not assess blinding | – | – | ST: −0.34 (*p* = 0.294) LT: −0.23 (*p* = 0.516) |
| *McMillan, Nolan & Kelly (1997)* | 20 | NR | P: Non-TrP SDN | Did not assess blinding | – | – | Immed: 0.35 (NSD) ST: 0.26 (NSD) |
| *Myburgh et al. (2012)* | 77 | 4 AG: 4 [2 non-compliant; 1 AE; 1 NR] | P: TrP SDN | Did not assess blinding | – | – | ST: −0.37 (NSD) |
| *Nabeta & Kawakita (2002)* | 34 | 7 AG: 2 [difficulty attending] SG: 5 [difficulty attending] | NP: Custom (blunt needle) | AG: 0.41 (0.01–0.81) SG: −0.18 (−0.62–0.26) Correct/Random | NSD between groups (*p* = 0.74) AG: 11/17 correct; 2/17 DK SG: 6/17 correct; 2/17 DK | Success | Immed: −0.12 (NSD) ST: −0.31 (NSD) B: −0.25 (NSD) |
| *Pecos-Martín et al. (2015)* | 72 | 0 | P: Non-TrP DN | Did not assess blinding | – | – | ST: −1.59 (*p* <0.001) LT: −1.93 (*p* <0.001) |
| *Tekin et al. (2013)* | 39 | 7 AG: 1 [ceased Ix] SG: 6 [ceased Ix] | NP: Custom (blunt needle) | Did not assess blinding | – | – | Immed: −0.88 (*p* = 0.034) ST: −1.62 (*p* = 0.000) |
| *Tough et al. (2009)*/*Tough et al. (2010)* | 41 | 7 AG: 3 [LTFU] SG: 4 [LTFU] | NP: Custom (blunt needle) | AG: 0.53 (0.30–0.75) SG: −0.67 (−0.93–−0.40) Correct/Incorrect | NSD between groups (*p*>0.2) AG: 10/19 correct; 9/19 DK SG: 1/18 correct; 4/18 DK | Success | ST/B: 0.11 (NR) LT: −0.61 (*p* = 0.67) |
| *Tsai et al. (2010)* | 35 | 0 | P: TrP SDN | Did not assess blinding | – | – | Immed: −0.91 (*p* <0.05) |

**Notes.**

[a] Dropouts for pain outcome.

[b] *Itoh & Katsumi (2005)* and *Itoh et al. (2007)* each had two eligible sham groups; in both of these trials one group had a non-penetrating (NP) sham and the other had a penetrating (P) sham (labelled accordingly in the first column of the table).

[c] *Itoh & Katsumi (2005)* only reported the number of participants from each group who guessed they were in the active group, therefore, to calculate the BI it was conservatively assumed that the remaining participants guessed they were in the sham group (i.e., no DK responses).

[d] Data not reported as mean/SD (could not calculate SMD).

n, number of participants (analysed for pain outcome)]; 95% CI, 95% Confidence Interval; AG, Active Group; SG, Sham Group; SMD, Standardised Mean Difference; −ve, Negative; Ax, Assessment; Ix, Intervention; NP, Non Penetrating; NSD, No Significant Difference; CEQ, Credibility/Expectancy Questionnaire; ST, Short-Term (24 hours to four weeks, closest assessment to one week); LT, Long-Term (one to six months, closest assessment to three months); Immed, Immediately post-intervention (<24 hours after first/only intervention); B, time-point at which Blinding was assessed; NR, Not Reported; DNR, Did Not Respond (to intervention); AE, Adverse Effects; P, Penetrating; TrP SDN, Superficial Dry Needling above Trigger Point; LTFU, Loss To Follow Up; DK, Don't Know; Non-TrP SDN, Superficial Dry Needling away from Trigger Point; Non-TrP DN, Dry Needling away from Trigger Point; AUC, Area Under Curve; GA, General Anaesthesia; SA, Spinal Anaesthesia.

Shading represents adequately blinded trials (based on critical appraisal criteria for review question 2).

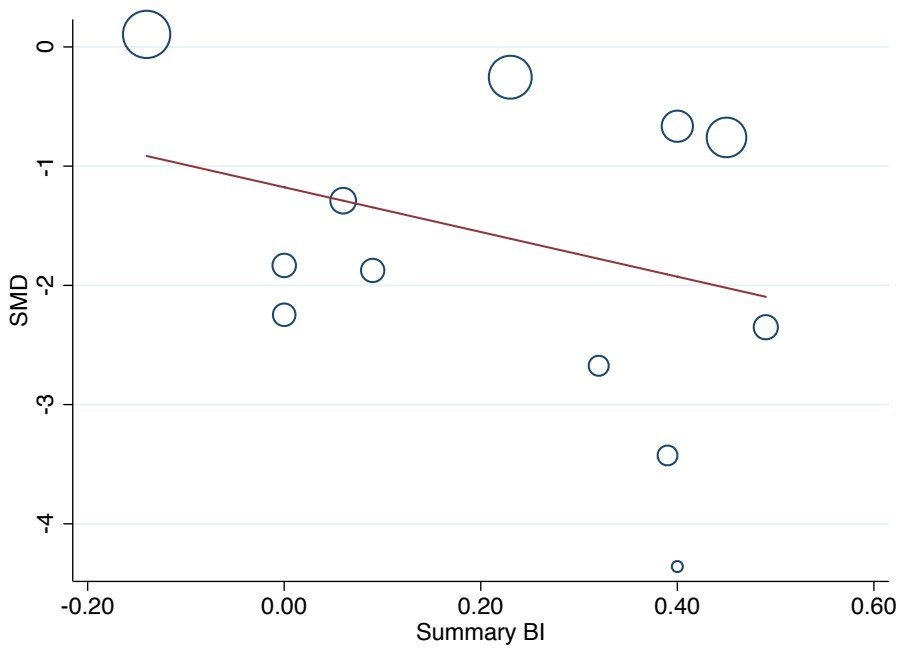

**Figure 3** **Bubble plot (meta-regression) of the influence of the summary BI (blinding effectiveness) on between-group effect size (pain) for pain assessments closest to the time point blinding was assessed ($N = 12$ group comparisons).** Each bubble represents one group comparison, and the size of each bubble is proportional to weight (inverse variance). Negative values for SMD are in favour of active dry needling. SMD, Standardised Mean Difference (effect size); BI, Blinding Index.

### Immediate intervention effect (<24 h after the first/only intervention)

There were seven group comparisons where immediate pain outcomes were collected (Fig. 4). One group comparison ($n = 31$) met the requirements for adequate blinding (*Inoue et al., 2006*), and intervention effects were statistically significant in favour of active dry needling (SMD −0.76 (95% CI [−1.49−−0.03])). For inadequately blinded group comparisons ($N = 6$; $n = 206$), there was no evidence of a difference in intervention effects between active and sham groups [pooled SMD -0.47 (95% CI -0.95 to 0.02)]. There was evidence of significant and substantial heterogeneity in the pooled group comparisons (*Higgins & Green, 2011*) (Fig. 4).

### Short-term intervention effect (24 h to one month, closest assessment to one week)

There were 20 group comparisons where short-term pain outcomes were collected (Fig. 5). For adequately blinded group comparisons ($N = 3$; $n = 122$) there was no evidence of a difference in intervention effects between active and sham groups (pooled SMD −0.40 (95% CI [−0.96–0.15])), whereas inadequately blinded group comparisons ($N = 17$; $n = 504$) had statistically significant intervention effects that favoured active dry needling (pooled SMD -0.71 (95% CI [−1.05−−0.38])). There was evidence of statistically significant and substantial heterogeneity of pooled group comparisons for the inadequately blinded

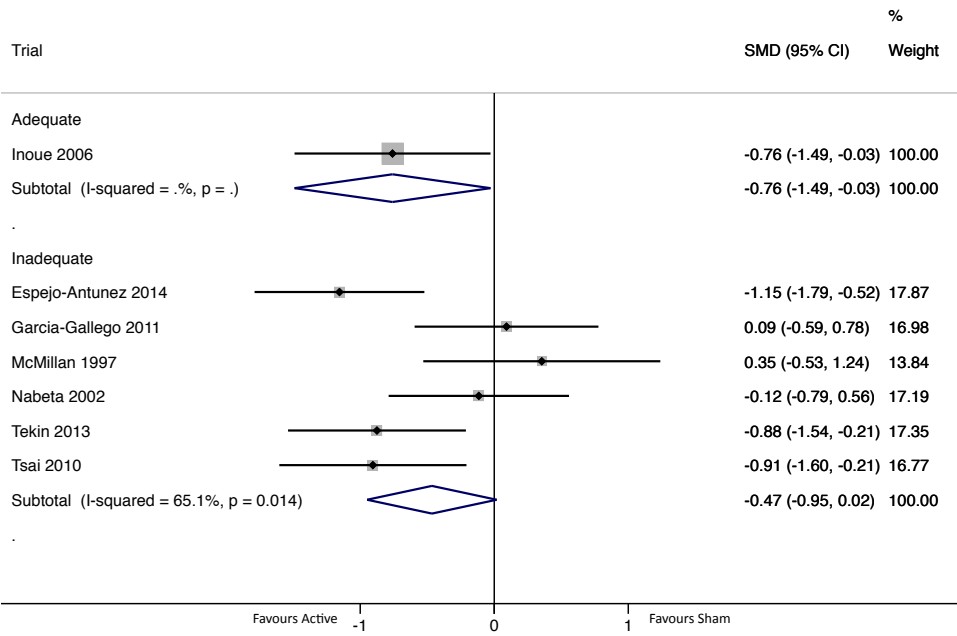

**Figure 4** Forest plot of pooled between-group effect sizes (pain) based on blinding adequacy, for pain assessments immediately after the first/only intervention (<24 h; N = 7 group comparisons).

subgroup, whereas the adequately blinded subgroup had moderate heterogeneity that was not significant (*Higgins & Green, 2011*) (Fig. 5).

### Long-term intervention effect (one to six months, closest assessment to three months)

There were 16 group comparisons where long-term pain outcomes were collected (Fig. 6). For adequately blinded group comparisons (N = 4; n = 202) there was no evidence of a difference in intervention effects between active and sham groups (pooled SMD −0.30 (95% CI [−0.62–0.02])), whereas inadequately blinded group comparisons (N = 12; n = 284) had statistically significant intervention effects that favoured active dry needling (pooled SMD -1.14 (95% CI [−1.64−−0.65])). There was evidence of statistically significant and substantial heterogeneity of pooled group comparisons for the inadequately blinded subgroup, whereas the adequately blinded subgroup had low heterogeneity that was not significant (*Higgins & Green, 2011*) (Fig. 6).

## DISCUSSION

### Key findings

This review aimed to determine whether blinding effectiveness and/or blinding adequacy moderated pain outcomes in dry needling trials. Of the 23 trials included in the meta-analyses, only 10 (43.5%) reported data that were sufficient to calculate the BI, and only five (21.7%) reported adequate blinding procedures. The small number and size of included

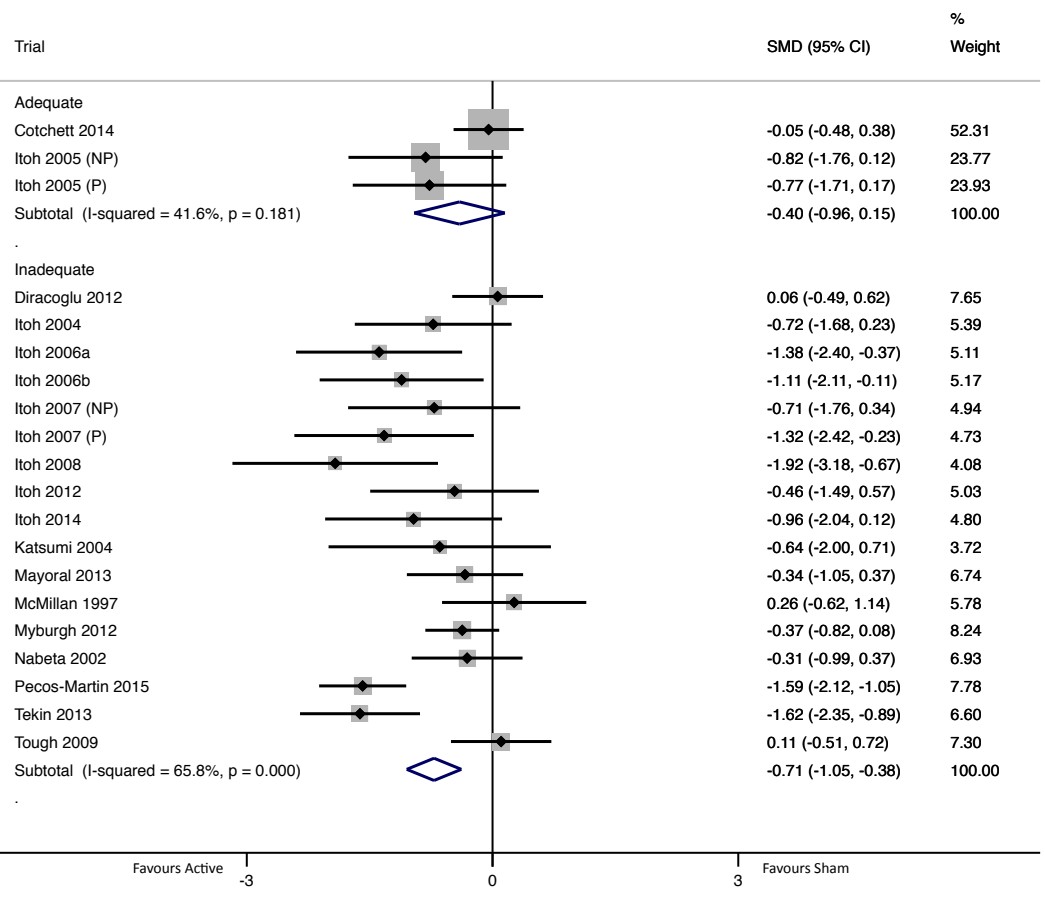

**Figure 5  Forest plot of pooled between-group effect sizes (pain) based on blinding adequacy, for pain assessments in the short-term (24 h to one month; *N* = 20 group comparisons).** Note: *Itoh & Katsumi (2005)* and *Itoh et al. (2007)* each had two eligible sham groups; in both of these trials one group had a non-penetrating (NP) sham and the other had a penetrating (P) sham (labelled accordingly in the figure).

trials meant that there was insufficient evidence to determine if a moderating effect of blinding effectiveness or adequacy existed (*Button et al., 2013*; *Higgins & Green, 2011*).

### Review question 1: Does blinding effectiveness moderate intervention effect on pain?

Blinding effectiveness was determined based on participant beliefs about whether they received active or sham dry needling. Table 5 presents the hypothesised moderation effect of the nine possible blinding scenarios on pain outcomes (adapted from *Bang et al. (2010)*), and the number of group comparisons in this review that fell into those scenarios. In this hypothesis, within each group (active or sham), intervention benefits would increase as more participants believe they received active dry needling (↑; Table 5). Theoretically, effective blinding would exist in Scenarios 4, 5, or 6, where intervention beliefs were approximately balanced between groups (shaded in Table 5). In contrast, the imbalance in active and sham groups in Scenarios 1–3 would favour the sham group and in 7–9 would favour the active group.

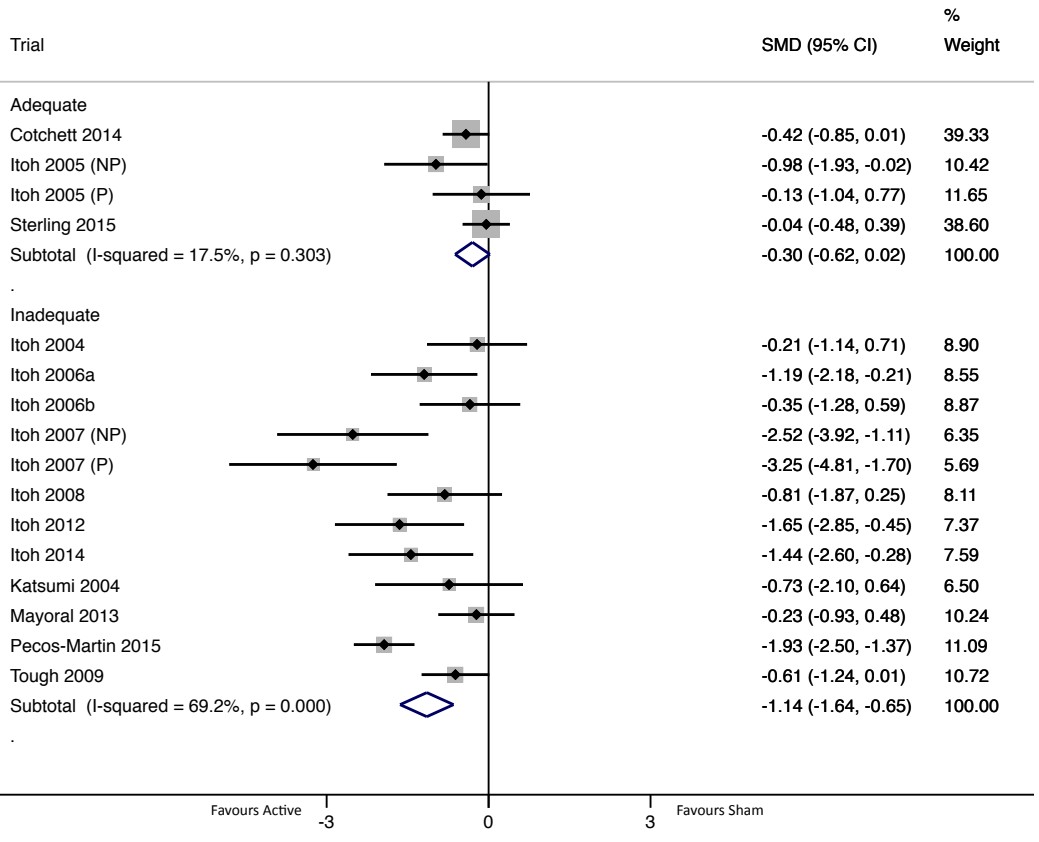

**Figure 6** **Forest plot of pooled between-group effect sizes (pain) based on blinding adequacy, for pain assessments in the long-term (one to six months; $N = 16$ group comparisons).** Note: *Itoh & Katsumi (2005)* and *Itoh et al. (2007)* each had two eligible sham groups; in both of these trials one group had a non-penetrating (NP) sham and the other had a penetrating (P) sham (labelled accordingly in the figure).

It was hypothesised that unbalanced beliefs between active and sham groups would moderate between-group differences in pain. No evidence of a moderating effect of the summary BI on pain was found, but the analysis may have been underpowered to detect it, as too it may have been underpowered to confidently conclude against it. Only 12 group comparisons were included in the analysis, marginally more than the minimum recommended number ($N = 10$) for a meta-regression (*Higgins & Green, 2011*). The current findings are in contrast to previous studies where significant associations between intervention outcomes and beliefs about allocation have been demonstrated (*Baethge, Assall & Baldessarini, 2013*; *Dar, Stronguin & Etter, 2005*; *McRae et al., 2004*), including pain outcomes in acupuncture trials (*Bausell et al., 2005*; *Vase et al., 2013*; *White et al., 2012*). Intervention effects favoured active dry needling irrespective of whether intervention beliefs were balanced between groups; this finding is consistent with *Moroz et al. (2013)* who found that the majority of acupuncture and dry needling trials reported positive outcomes, regardless of blinding effectiveness. The findings of the meta-regression are likely to have been threatened by the inclusion of underpowered trials (*Button et al., 2013*), by other

**Table 5 Hypothesised effects of intervention belief on pain for group comparisons where the Blinding Index (BI) could be calculated ($N = 12$ group comparisons) (adapted from *Bang et al. (2010)*).** Shading represents theoretically effective blinding scenarios (i.e. intervention beliefs approximately balanced between active and sham groups).

| No. | AG beliefs | SG beliefs | Hypothesised moderation effect of intervention belief on pain outcomes | | | N (%) | n (%) |
|---|---|---|---|---|---|---|---|
| | | | AG | SG | Between group | | |
| 1 | Incorrect (sham) | Incorrect (active) | ↓ | ↑ | Large; in favour of SG | 0 (0) | 0 (0) |
| 2 | Random | Incorrect (active) | – | ↑ | Small; in favour of SG | 0 (0) | 0 (0) |
| 3 | Incorrect (sham) | Random | ↓ | – | Small; in favour of SG | 0 (0) | 0 (0) |
| 4 | Incorrect (sham) | Correct (sham) | ↓ | ↓ | None (reduced in both groups) | 0 (0) | 0 (0) |
| 5 | Random | Random | – | – | None | 0 (0) | 0 (0) |
| 6 | Correct (active) | Incorrect (active) | ↑ | ↑ | None (inflated in both groups) | 8 (67) | 145 (58) |
| 7 | Correct (active) | Random | ↑ | – | Small; in favour of AG | 3 (25) | 72 (29) |
| 8 | Random | Correct (sham) | – | ↓ | Small; in favour of AG | 0 (0) | 0 (0) |
| 9 | Correct (active) | Correct (sham) | ↑ | ↓ | Large; in favour of AG | 1 (8) | 31 (13) |

**Notes.**

No., Scenario Number; AG, Active Group; SG, Sham Group; N, Number of group comparisons; n, number of participants.

threats to internal validity (e.g., biases associated with therapist expectation (*Gracely et al., 1985*) and the high likelihood of publication bias (Fig. 2A)), and by substantial statistical heterogeneity and confounding due to the non-randomised nature of the meta-regression analysis (*Higgins & Green, 2011*). Therefore, further research is needed to quantify a moderating effect of blinding effectiveness on pain outcomes.

Inconsistent techniques and incomplete reporting of blinding assessments make it difficult to draw robust conclusions. Overall, 14 trials (58%) in this review reported some form of blinding effectiveness or intervention credibility data, which is markedly greater than in previous samples (e.g., between 2–8% of random samples of clinical trials reported assessments of blinding (*Fergusson et al., 2004*; *Hróbjartsson et al., 2007*)). However, of these 14 trials, only 10 reported data that were sufficient to calculate the BI. Where reported blinding data were insufficient to calculate the BI, authors were contacted to request the raw data but authors either did not respond or no longer had access to the data. Given the strong motivation to report success, there is a possibility of underreporting when blinding assessments indicate ineffective blinding (i.e., reporting bias) (*Hróbjartsson et al., 2007*).

The lack of data to confirm the influence of blinding effectiveness on trial outcomes means that currently blinding 'success' is largely subjective (*Bang et al., 2010*). This is evidenced by the universal author conclusion of blinding 'success' (where reported), despite varied patterns in the blinding data (Table 4).

### Review question 2: Does blinding adequacy moderate intervention effect on pain?

It was hypothesised that inadequate blinding procedures would exaggerate intervention effects. Threats to the internal validity of included trials, coupled with the limitations of meta-analytical techniques precluded definitive conclusions. For immediate assessments, there was no evidence of a difference in intervention effects between adequately and inadequately blinded group comparisons (Fig. 4), but drawing inferences from this finding

is difficult due to the small sample of group comparisons ($N = 7$; with only $N = 1$ adequately blinded). However, with the caveats of small samples, generally unclear RoB, and the limitations of subgroup analyses, the available evidence suggests that inadequate blinding procedures could lead to exaggerated intervention effects in dry needling trials in the short-term and long-term.

In the short-term and long-term, there were statistically significant intervention effects in favour of active dry needling for inadequately blinded group comparisons, whereas adequately blinded group comparisons showed no difference between groups (Figs. 5 and 6). Differences in pooled pain outcomes between inadequately and adequately blinded group comparisons were moderate to large (short-term difference in SMD = 0.31; long-term difference in SMD = 0.84), and in the long-term there was no overlap of pooled 95% CIs (i.e., significance guaranteed at $p < 0.05$). In addition, in both the short-term and long-term, the adequately blinded group comparisons had more statistically homogenous results, and in the long-term the 95% CI for adequately blinded comparisons was also more precise despite having fewer group comparisons. These findings together suggest that inadequate blinding might be associated with greater heterogeneity and lower precision in group comparisons.

The current findings are consistent with the findings of previous meta-analyses investigating moderating effects of inadequate blinding procedures (*Hróbjartsson et al., 2014*; *Savović et al., 2012*). More specifically, exaggeration of intervention effects has been found in trials with inadequate allocation concealment and/or outcome assessor blinding (*Hróbjartsson et al., 2012*; *Hróbjartsson et al., 2013*; *Jüni, Altman & Egger, 2001*; *Nüesch et al., 2009*; *Schulz et al., 1995*; *Wood et al., 2008*), and these two domains were the only determinates of blinding adequacy in the current review because all included trials demonstrated adequate participant blinding and no trials attempted to blind therapists.

Lack of adequate blinding procedures means that at present, specific effects of dry needling cannot be distinguished from effects due to bias. Blinding of therapists and research personnel was either not attempted or poorly reported by all included trials (Table 3). There are clearly substantial practical challenges with therapist blinding, however, potential effects of non-blinded therapists (*Cook et al., 2013*; *Gracely et al., 1985*; *Moher et al., 2010*; *Savović et al., 2012*; *Vase et al., 2015*) warrants research in this direction. Acupuncture studies have attempted therapist blinding using custom-made sham needle devices (*Takakura et al., 2010*; *Takakura & Yajima, 2007*), which may have potential for application in future dry needling trials. Blinding of research personnel should be a relatively simple procedure and needs greater attention and/or clearer reporting. Participant attrition, another major source of potential bias, should be accounted for using statistical methods such as intention-to-treat analysis using multiple imputation, possibly with adjustments for informative missingness, or adjustments based on covariates. In addition, there were more dropouts in sham groups due to 'no response to intervention' (where reported: $n = 15$ in sham groups versus $n = 4$ in active groups; Table 4), which could have contributed to biases favouring active dry needling.

## Strengths and limitations

The strengths of this review included prospective peer review and registration of the protocol (PROSPERO), adherence to the PRISMA statement for reporting (*Moher et al., 2009*), and independent screening for trial eligibility, data extraction, and RoB assessments by two reviewers. The search strategy was comprehensive and trials were not limited to the English language. Despite attempts to limit the impact of publication bias on the current results by searching trial registrations and thesis databases, asymmetry of funnel plots suggests publication bias was present (Fig. 2). The small number of trials in one of the funnel plots (<10; Fig. 2B) meant that it could not be confidently interpreted (*Egger et al., 1997*; *Higgins & Green, 2011*).

The current findings should be interpreted with caution. The meta-analytical techniques used in this review are not randomised comparisons and are therefore observational in nature (*Higgins & Green, 2011*). The strength of inferences is therefore limited by potential confounding by uncontrolled covariates, and subgroups may have differed in capacity to detect effects (*Higgins & Green, 2011*). However, for review question 2, the *a priori* hypothesis, the statistical significance of the findings, and the consistency of the difference across comparisons strengthen the validity of the inferences (*Oxman & Guyatt, 1992*). That no studies to date have made head-to-head comparisons of blinded versus non-blinded dry needling interventions precludes any further analysis, aside from indirect comparisons. The meta-analyses used random effects modelling to allow for statistical heterogeneity between trials (*Higgins & Green, 2011*). Homogeneity was improved for meta-analyses investigating blinding adequacy ($I^2$ values <75%; Figs. 4–6), in which comparisons were grouped based on four RoB domains, for three pre-defined time periods, so these analyses may be more reliable (*Higgins & Green, 2011*).

The included trials were methodologically heterogeneous, and many were likely to have had a high risk of null findings in the presence of small to moderate effects due to insufficient power ($N = 20$ group comparisons with $n < 50$ participants, with power clearly achieved for the pain outcome in only three trials, and zero for blinding assessment outcomes) (*Button et al., 2013*). Trials were also clinically diverse in terms of participant health condition, pain chronicity, age, and intervention dose, which may have confounded results, in particular because the aetiology of pain may influence the specific effects of dry needling (*Cagnie et al., 2013*; *Dommerholt, 2011*), as well as non-specific effects (*Tracey, 2010*). The limited number of trials precluded investigation of potential covariates (i.e., sensitivity or multivariable meta-regression analyses) (*Higgins & Green, 2011*).

Contrary to best practice, active group data were used twice in several meta-analyses that included trials with two eligible sham groups (*Itoh & Katsumi, 2005*; *Itoh et al., 2007*) (Figs. 3, 5 and 6), which may have caused unit-of-analysis errors due to correlations between the non-independent comparisons (*Higgins & Green, 2011*). However, due to extremely small sample sizes ($n = 8$ to 10 participants in the relevant active groups) and potential differences in physiological effects of the sham interventions (i.e., penetrating versus non-penetrating) (*Lund, Näslund & Lundeberg, 2009*), it was decided that the active group data could not be split, nor could the sham groups be combined as recommended by *Higgins & Green (2011)*.

To determine whether the current review required updating (original search completed in February 2016), a citation search was undertaken for trials included in the current systematic review (292 citations since January 2016 as at 18th of September 2017, with reference lists of 39 potentially relevant systematic reviews also reviewed). This search revealed 47 new prospective primary studies of dry needling; of these, only one was blinded using sham dry needling and this trial did not report an assessment of blinding effectiveness (*Mason et al., 2016*). Addition of one trial to the current review was unlikely to significantly alter the results, therefore the review was not updated (*Elkins, 2018*).

## CONCLUSIONS

This review found insufficient data to understand moderating effects of blinding effectiveness or adequacy on pain; therefore recommendations about interpreting trial outcomes with reference to blinding are premature. However, consistent with previous reviews, the current review found a bias in favour of active dry needling when trials were inadequately blinded for short-term and long-term pain outcomes. Due to the limitations of subgroup comparative analyses and threats to the validity of the included trials (particularly insufficient power), the findings of this review should be interpreted with caution. We did not aim to determine whether or not dry needling is superior to sham, but we can confidently conclude that should researchers propose further trials in this or related areas, they should be adequately blinded and collect robust blinding data.

## ACKNOWLEDGEMENTS

The authors would like to thank Dr Tasha R Stanton for her valuable assistance in the preparation of this manuscript, and Dr. Beben Benyamin and Dr Terry Boyle for their statistical support with the meta-analytical techniques.

### Funding

Felicity A Braithwaite and Lok Sze Katrina Li were each supported by an Australian Government Research Training Program Scholarship. G Lorimer Moseley was supported by a Principal Research Fellowship from the National Health and Medical Research Council of Australia (NHMRC) ID 1061279. The funders had no role in study design, data collection and analysis, decision to publish, or preparation of the manuscript.

### Grant Disclosures

The following grant information was disclosed by the authors:
Australian Government Research Training Program.
National Health and Medical Research Council of Australia (NHMRC): 1061279.

### Competing Interests

G Lorimer Moseley has received support from Pfizer, Kaiser Permanente, Providence Healthcare, Agile Physiotherapy, Results Physiotherapy, Workers' Compensation Boards

in Australia, Europe and North America, the International Olympic Committee and the Port Adelaide Football Club. G Lorimer Moseley receives royalties for several books on pain and speaker's fees for talks on pain, physiotherapy, and rehabilitation.

## Author Contributions

- Felicity A. Braithwaite conceived and designed the experiments, performed the experiments, analyzed the data, contributed reagents/materials/analysis tools, prepared figures and/or tables, authored or reviewed drafts of the paper, approved the final draft.
- Julie L. Walters, Marie T. Williams and Maureen P. McEvoy conceived and designed the experiments, performed the experiments, contributed reagents/materials/analysis tools, prepared figures and/or tables, authored or reviewed drafts of the paper, approved the final draft.
- Lok Sze Katrina Li performed the experiments, contributed reagents/materials/analysis tools, authored or reviewed drafts of the paper, approved the final draft.
- G. Lorimer Moseley conceived and designed the experiments, performed the experiments, contributed reagents/materials/analysis tools, authored or reviewed drafts of the paper, approved the final draft.

## Data Availability

The research in this article did not generate any raw data or code because this was a systematic review of published literature.

## Supplemental Information

Supplemental information for this article can be found online at http://dx.doi.org/10.7717/peerj.5318#supplemental-information.

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
