# Peer review of "Effectiveness and adequacy of blinding in the moderation of pain outcomes: Systematic review and meta-analyses of dry needling trials"

_PeerJ, doi:10.7717/peerj.5318_

## Round 0.1 · original submission · Minor Revisions

As the reviewers have noted, and I agree, there is considerable merit in this manuscript and relatively few issues to address. The reviewers have suggested a few small edits and requests for clarification (e.g., always including abbreviations in full at their first use) which I ask you to consider and respond to in each case.

I will add a small number of additional points myself here:

1. Care needs to be taken when making statements such as "no significant moderating effect" (Line 40) as there is the potential for confusion between statistical significance and clinical/practical significance. I would suggest avoiding unqualified instances of "significance", "significant", or "significantly".

2. Related to the above, statements of no effect should be avoided (outside of equivalence studies) as absence of evidence is not evidence of absence. The "no difference" on Line 42 would be clearer as "no evidence of a difference". See also Lines 346 and 363 (and this should be kept in mind when editing Line 40).

3. The use of strict inequalities (Lines 151 and 182) could be avoided by using greater than or equal to signs instead. This avoids the potential readers’ question of whether exactly 18 or exactly 75% meets the respective criterion.

4. I'm not convinced that "chance" is a risk of bias (Line 196). I'm assuming this is based on Button, et al. (2013). A bias leads to a systematic under- or over-estimation of an effect with the systematic component being crucial. To quote from Porta’s Dictionary of Epidemiology (page 21) "BIAS Systematic deviation of results or inferences from truth. Processes leading to such deviation. An error in the conception and design of a study—or in the collection, analysis, interpretation, reporting, publication, or review of data—leading to results or conclusions that are systematically (as opposed to randomly) different from truth." The points raised in Button, et al., from my reading, are based around publication effects or by looking only at statistically significant effects, which are assessed in your manuscript separately. Under-powered studies will not on average produce higher or lower effect sizes than adequately or over-powered studies and so are not biased in the usual sense of the term. I don't think anything would be lost here if this additional RoB item was removed, or perhaps it could be retained under a different heading? Note that this point arises again at Lines 433-434 where this could apply to publication bias but not strictly from study power alone.

5. Cohen's "large" effect size is 0.8 not 0.7 as stated on Line 221 (see Cohen, 1988 page 40 or https://en.wikipedia.org/wiki/Effect_size#Cohen's_d if you do not have access to a copy).

6. Consistent decimal places should be used for all p-values and I would strongly suggest 3 dp for all but very small p-values where "<0.001" should be used. (See Line 31 1 for a mixture of 2 and 3 decimal places.) Note that Line 348 ("p=0.00") should also use 3 decimal places or "<0.001".

7. Type 2 errors, like Type 1 errors, can never be ruled out (Lines 423-424) and I'd suggest rewording this to make it clear that the analyses may have been underpowered to detect clinically significant effects (if you agree with that interpretation of course).

8. The use of the overlap rule for 95% CIs (Line 473) is problematic as this is equivalent, depending on the specific test, to a two-sided test at around the 0.01 level and some overlap is possible for significance at the 0.05 level. While this doesn't alter the interpretation here, it would be helpful to reword this sentence to make it clear that non-overlapping 95% CIs guarantees significance at the 0.05 level rather than being a necessary condition.

9. Attrition is indeed a source of bias (Lines 502-503) but not one that can be overcome with larger samples as seems to be suggested here. The effects of informative missingness, and MAR when the necessary covariates are not conditioned on, are independent of sample size.

Well done on producing such a clear and focused manuscript. I look forward to seeing more articles come out of this project.

Reviewer 1 ·

Basic reporting

Well done

Experimental design

Well done

Validity of the findings

Reasonably well done given available evidence and validated/acceptable methods

Additional comments

Authors performed a very thorough and rigorous review. Specific, mostly minor, comments are:
1. What is inverse-var random effect model? DL model?
2. Blinding adequacy is 1/0?
3. Some parts are repetitive. I think shorter text is more read by readers.
4. Blinding is relevant in n-of-1?
5. Line 328: how to determine intervention credibility?
6. Baethge et al. (Psychother Psyochsom2013) may be cited; a similar research was done but without BI.
7. Effect estimate is Effect size? Please clarify how it was defined. Change score or difference in change scores or something else?
8. Line 378: P-value or CI may be added?
9. Line 437 and 479: randomization is important in meta reg and subgroup?
10. Line 481: Clinical significance is addressed?

Editorial:
1. Line 123: The vs. the.
2. Line 325: What is DN?
3. Line 414: broken letter.
4. Line 524-5: N and n is correct?
5. Line 551: pearled?
6. Table 2: 0.2.0.

·

Basic reporting

The English grammar and spelling were correct with adequate and appropiate scientific terms. The literature references are update and according to the methods description. The background and context are correct and justify the research question and hypothesis, which are very appropiate and necessary. The article structure, figures, and tables showed a well-planned study which provides important considerations in a correct way.

Experimental design

The experimental research design of this manuscript shows a high quality research in a rigorous way. Methods are described in a clear style with enough detail. Nevertheless, I only suggest adding the Ethics committee approval code in the methods section, which is an important item in order to add this information to the PRISMA and PROSPERO data.

Validity of the findings

The validity of the findings is appropiate. The conclusion are established according to the objectives. Therefore, the finddings are useful and necessary according to the described literature.

Additional comments

Congratulations for the authors. This manuscript is well-planned study and provides important information which is necessary in the dry needling field. The placebo and sham treatments mechanisms needs to be investigated and this research presents the necessity of futher investigations regarding the effects of these interventions in order to establish the adequate effectiveness in blinded randomized clinical trials. I only suggests that the Ethics committee approval code should be added in order to clarify this key point in the methods section. In addition, adding the MeSH term "Myofascial pain syndromes" may be useful.

·

Basic reporting

All satisfactory

Experimental design

All satisfactory

Validity of the findings

All satisfactory

Additional comments

The most fascinating part of this report, for me, was the revelation that out of 4894 individual publications only 24 were good enough to be included in the analysis. And that, of these, 19 trials showed risk of methodological bias that was high or unclear. Five trials were adequately blinded, and blinding was assessed and sufficiently reported to compute the blinding index in 10 trials.

This is a damning comment on the quality of acupuncture research, and well worth publishing.
The only bit that I would change is the conclusion:

"Future trials should be adequately blinded and should collect robust blinding data to allow quantification of moderating effects of blinding effectiveness on pain outcomes; "

My conclusion, from the same data would be that it isn't worth spending any more time and money on acupuncture research. After over 4000 trials there is still room for doubt about whether it works to any useful extent. and the best quality trials show that either it doesn't work at all, or produces a benefit that's too small to be useful to the patient, my conclusion would be that no further research is needed. Another 4000 trials are unlikely to change the conclusions.

---

## Round 0.2 · Minor Revisions

Reviewer 1 has a query about the wording of the sentence on lines 220-221. Speaking as a biostatistician, I would probably have written something close to the original (“The Restricted Maximum Likelihood (REML) model was used to estimate between-trial variance.”), although I’d favour “approach” or “method” over “model” if such a noun was to be included there, and probably use slightly fewer words: “Restricted Maximum Likelihood (REML) was used to estimate the between-trial variance.” I will leave this point to the authors’ discretion as I think the current phrasing is acceptable if they prefer to retain it.

Reviewer 2 has indicated that they are happy with this version of the manuscript.

Reviewer 3 has sent a strong indication with their previous recommendation of minor revisions becoming one to reject. I think that there is an extremely important distinction between recommendations around assessing and reporting blinding (which is clearly a good thing and one that ought to be done much more in many areas of research) and recommendations around pursuing further research in the area of dry needling given its evidence base.

The text noted by Reviewer 3 currently reads:

"Future trials should be adequately blinded and should collect robust blinding data to allow quantification of moderating effects of blinding effectiveness on pain outcomes; these will be important steps towards determining the true mechanisms of dry needling."

I can read this in one of two ways, that any future trials should be adequately blinded, etc., or that there should be future trials in this area. The call for further research in the second sense is often the default ending to a manuscript (with it being banned by several journals and common enough to have a Wikipedia page devoted to it: https://en.wikipedia.org/wiki/Further_research_is_needed).

I feel that focusing on the methodological aspect is entirely appropriate, and this may be the intention here. If so, the methodological aspect could be made clearer, and more general, by replacing the above text with something like:

"Any future trials in this or related areas should be adequately blinded and collect robust blinding data to allow quantification of moderating effects of blinding effectiveness on pain outcomes."

Then rather than the clichéd “more research is needed,” reading of this text, the manuscript clearly ends on an important methodological point, namely that without adequate blinding and an assessment of potential biases from participants being unblinded, disentangling causal evidence from RCTs in pain management is, at best, fraught.

I have also made a small number of relatively minor suggestions listed below:

1) Line 62. Apologies for the pedantry, but on re-reading the manuscript, this time around I wondered why trial staff were last in this list and considered that perhaps this list was intended to be ordered from highest to lowest (subjective) risk. Otherwise, perhaps a temporal ordering would work? (trial staff [assuming they are involved in screening and allocation], therapists, recipients, outcome assessors, data analysts?)

2) Line 94. Just a suggestion, but perhaps a comma after “trials”.

3) Line 480. Sorry for not picking this up before, but on re-reading I wondered exactly what it was about the random effects that you were potentially attributing to inadequate blinding and assumed that this was their variability. One option for wording to make this clearer would be: “These findings together suggest that inadequate blinding might be associated with greater heterogeneity and lower precision in group comparisons.”

4) Line 502. I wonder if adding “using multiple imputation, possibly with adjustments for informative missingness,” after “intention-to-treat analysis” might not be useful to the reader in making it clear what these “analyses” might be (strictly, ITT refers to the research question rather than the analyses themselves although it is not unusual to hear people talk of “intention to treat analyses” or “per-protocol/efficacy analyses”).

5) Line 532. The “chance findings” here, I’m assuming refers to false negative chance results and that could be made clearer, perhaps “high risk of null findings in the presence of small to moderate effects due to insufficient power”. A two group comparison with the usual 80% power, two-sided 0.05 significance level would require n=26/group to detect a 0.8 SD difference, n=64/group to detect a 0.5 SD difference, and n=394/group to detect a 0.2 SD difference.

6) Line 539. I appreciate that multivariate is often used where multivariable ought to be used, but I wonder if it would be worth using the “more correct” “multivariable meta-regression” here if this is referring to the right-hand side of the equation, where covariates could have been added? (See https://www.ncbi.nlm.nih.gov/pmc/articles/PMC3518362/).

Reviewer 1 ·

Basic reporting

None

Experimental design

None

Validity of the findings

None

Additional comments

One small comment:

I think "REML model was used.." is an incorrect sentence. Authors have statistical collaborators or coauthors. So they should consult with statistician and fix the sentence if they agree.

·

Basic reporting

Thanks for addressing the review suggestions

Experimental design

Thanks for addressing the review suggestions

Validity of the findings

Thanks for addressing the review suggestions

Additional comments

Thanks for addressing the review suggestions

·

Basic reporting

OK

Experimental design

OK

Validity of the findings

The findings are OK. The conclusions are totally wrong, in my opinion.

Additional comments

I made only one suggestion but the authors declined to implement it. Therefore I think that this paper should be rejected.
I said
"My conclusion, from the same data would be that it isn't worth spending any more time and money on acupuncture research. After over 4000 trials there is still room for doubt about whether it works to any useful extent. and the best quality trials show that either it doesn't work at all, or produces a benefit that's too small to be useful to the patient, my conclusion would be that no further research is needed. Another 4000 trials are unlikely to change the conclusions.’"

The authors' response was
"We agree that further research is unlikely to change the conclusions of acupuncture research."
They seem to agree but then declime to change their conclusions on the follwoing grounds.
"However, the current review investigated dry needling specifically and excluded acupuncture trials. The philosophy and the proposed mechanisms for dry needling are considered different from those of acupuncture"

Since one thing that has emerged from the 4000+ trials of acupuncture is that it makes no difference to the outcome whether needles are inserted in the traditional acupuncture points or anywhere else, these grounds seem to me to be invalid. It's already known that the position of the needles makes no difference.That's how we know that traditional acupuncture points are figments of imagination,, dating from a prescientific age.

If we accepted the authors argument, we'd have to accept another 4000 trials from every subtype of acupuncture -auricullar, electro etc etc etc.

---

## Round 0.3 · Minor Revisions

I sent the manuscript back to David for his thoughts, which I greatly appreciate him providing and continuing to provide. As you can see, my suggested wording has not completely alleviated his concerns. I'm personally comfortable that your article is not encouraging further research in acupuncture, which would not be a justified conclusion given your research question and results, but rather is making an important methodological point about studies in this and related areas. If anything, your results provide a potential mechanism for any positive results in open label or unintentionally unblinded pain trials.

I've sent this manuscript back to you with David's further thoughts as I am in agreement with David's 2013 article in Anesthesia & Analgesia, which notes that any benefits from acupuncture appear to be subclinical, and I imagine that other readers will also have strong feelings about your article. You are welcome to consider his thoughtful suggestion about an additional clause concerning further research in this area but I am not making these changes mandatory.

In fact, I see your manuscript as addressing the first half of a point from his 2013 article: "The acupuncture and no-acupuncture groups were, of course, neither blind to the patients nor blind to the practitioner giving the treatment. It is not possible to say whether the observed difference is a real physiological action or whether it is a placebo effect of a rather dramatic intervention. Though it would be interesting to know this, it matters not a jot, because the effect just is not big enough to produce any tangible benefit." I read your manuscript as consistent with the second point there as well, it provides an interesting study of blinding's role in observed effects and does not make claims about acupuncture’s effectiveness or call for further research on this topic (nor does it do the opposite, which is David's point, but then again, that was not its purpose).

I note that Anesthesia & Analgesia published another study including acupuncture in December 2017 (Lin, et al.), which in its accompanying infographic states: "Strong evidence for acupuncture as a complementary treatment for chronic pain that has been shown to decrease the usage of opioids was found." I will also note the last few sentences from this article’s abstract: "Overall, weak positive evidence was found for yoga, relaxation, tai chi, massage, and manipulation. Strong evidence for acupuncture as a complementary treatment for chronic pain that has been shown to decrease the usage of opioids was found. Few studies were found in which integrative medicine approaches were used to address opioid misuse and abuse among chronic pain patients. Additional controlled trials to address the use of integrative medicine approaches in pain management are needed.” If Anesthesia & Analgesia, the 6th ranked journal in anaesthesiology out of 31 according to Web of Science, can continue to report on acupuncture, and even call for further research in that area—which is a step further than I would have been willing to go, I am reassured that my position, namely that the results in all studies must be allowed to stand in the absence of demonstrable methodological or ethical issues, is the correct one here, although I do also agree with David about the value of further research in this area.

Whether you choose not to make any revisions at this stage or to modify the wording in light of David's comments, which I will reiterate, reflects a reaction that I imagine will not be rare, I will recommend acceptance and the article will pass upwards to the section editor for their final consideration.

·

Basic reporting

All the authors have done in response to my comments is to replace

"Future trials should be adequately blinded and should collect robust blinding data to allow quantification of moderating effects of blinding effectiveness on pain outcomes; these will be important steps towards determining the true mechanisms of dry needling."

with

"Any future trials in this or related areas should be adequately blinded and collect robust blinding data to allow quantification of moderating effects of blinding effectiveness on pain outcomes’ .

Although this is slightly better, it still implies that the authors are unwilling to offer advice on future trials. The authors said after the first review that "We agree that further research is unlikely to change the conclusions of acupuncture research." Yet they have still not included this important statement in the paper. In my opinion, this is not scientific objectivity but an implicit condonement of pseudo-scientific medicine, Some very respectable journals like Anesthesia and Analgesia have stopped accepting papers on things like acupuncture altogether. One reason for that is that such papers tend to keep alive the idea that there is controversy. That is standard practice for alternative medicine advocates. It is exactly the same tactic as was used by the tobacco industry: manufactured doubt.

In my view the paper should be rejected if the authors are not willing to include something like
"Further research is unlikely to change the conclusions of acupuncture research, and to that extent no further research can be justified. If more trials are done in this or related areas should be adequately blinded and collect robust blinding data ..."

Experimental design

No comment

Validity of the findings

No comment

Additional comments

All the authors have done in response to my comments is to replace

"Future trials should be adequately blinded and should collect robust blinding data to allow quantification of moderating effects of blinding effectiveness on pain outcomes; these will be important steps towards determining the true mechanisms of dry needling."

with

"Any future trials in this or related areas should be adequately blinded and collect robust blinding data to allow quantification of moderating effects of blinding effectiveness on pain outcomes’ .

Although this is slightly better, it still implies that the authors are unwilling to offer advice on future trials. The authors said after the first review that "We agree that further research is unlikely to change the conclusions of acupuncture research." Yet they have still not included this important statement in the paper. In my opinion, this is not scientific objectivity but an implicit condonement of pseudo-scientific medicine, Some very respectable journals like Anesthesia and Analgesia have stopped accepting papers on things like acupuncture altogether. One reason for that is that such papers tend to keep alive the idea that there is controversy. That is standard practice for alternative medicine advocates. It is exactly the same tactic as was used by the tobacco industry: manufactured doubt.

In my view the paper should be rejected if the authors are not willing to include something like
"Further research is unlikely to change the conclusions of acupuncture research, and to that extent no further research can be justified. If more trials are done in this or related areas should be adequately blinded and collect robust blinding data ..."

---

## Round 0.4 · accepted · Accept

Thank you for considering David's thoughtful comments. I feel that your revision will help to remind the reader, should they need reminding, about your study's purpose and makes the methodological point very clear. Well done.

#